# Can participatory approaches strengthen the monitoring of cyanobacterial blooms in developing countries? Results from a pilot study conducted in the Lagoon Aghien (Ivory Coast)

Veronica Mitroi[1], Kouadio Chrislain Ahi[2], Pierre-Yves Bulot[1,3], Fulbert Tra[2], José-Frédéric Deroubaix[4], Mathias Koffi Ahoutou[5,6], Catherine Quiblier[7,8], Mariatou Koné[2], Julien Coulibaly Kalpy[5], Jean-François Humbert [1]*

1 INRAE-Sorbonne Université, iEES Paris, Paris, France, 2 Institut d'Ethnosociologie, Université Félix Houphouët Boigny, Abidjan, Côte d'Ivoire, 3 INRAE-LISIS, Cité Descartes, Marne-la-Vallée, France, 4 LEESU, École des Ponts ParisTech, Champs-sur-Marne, France, 5 Institut Pasteur d'Abidjan, Abidjan, Côte d'Ivoire, 6 Université Jean Lorougnon Guédé, Daloa, Côte d'Ivoire, 7 MNHN, UMR 7245 Molécules de Communication et Adaptation des Micro-organismes, Paris, France, 8 Université de Paris, Paris, France

* jean-francois.humbert@upmc.fr

## Abstract

Monitoring of cyanobacteria in freshwater ecosystems is a complex task, which is time consuming and expensive due to the chaotic population dynamics and highly heterogeneous distribution of cyanobacteria populations in water bodies. The financial cost constitutes a strong limitation for the implementation of long-term monitoring programs in developing countries, particularly in Africa. The work presented here was performed in the framework of an international project addressing the sustainable monitoring and management of surface water resources used for the production of drinking water in three African countries. We tested the potential of a citizen approach for monitoring cyanobacterial blooms, which are a growing threat to the drinking water supply. This pilot study was designed, implemented and evaluated in close interaction with the Pasteur Institute of the Ivory Coast and with the populations of three villages located on the shoreline of a freshwater lagoon located near Abidjan city. Based on the use of a smartphone application, the citizens of the three villages were invited to report water color changes, as these changes could reflect cyanobacteria proliferations. A two-year experimentation period has shown that it is possible to mobilize the local populations to monitor cyanobacterial blooms. The data collected by citizens were consistent with the data obtained by a classical monitoring of cyanobacteria performed over seven months, but it appeared that new approaches were needed to validate the citizen data. This participatory approach also provided great improvements to the understanding and awareness of local populations regarding water quality and cyanobacterial bloom issues. Finally, we discuss some of the difficulties and limitations of our participatory monitoring approach that should be considered by further implementations. Despite these difficulties, our work

**Data Availability Statement:** All data obtained by the citizen monitroing are available on the website: https://five.epicollect.net/project/wasaf.

**Funding:** This study was supported with funds from Fonds Français pour l'Environnement Mondial (FFEM).

**Competing interests:** NO authors have competing interests.

suggests that citizen monitoring is a promising approach that may complement the classical approach to sustainable monitoring of cyanobacteria in developing countries.

## Introduction

Monitoring the proliferations of potentially toxic cyanobacteria in freshwater ecosystems is an essential pillar for the assessment and management of environmental and sanitary risks associated with these phenomena. Numerous studies have therefore been conducted on this issue over the last twenty years. It has been shown that (i) one of the main difficulties in cyanobacteria monitoring is due to the temporal and spatial variability in their vertical and horizontal distributions in water bodies [e.g. 1–3], meaning that the choice of sampling strategy and the tools used for sampling have great impacts on the quality of the data collected [3, 4], and (ii) accurate monitoring of cyanobacteria and their associated risks is time consuming and expensive [5].

As a matter of consequence, many efforts have been made to develop and validate new innovative devices for the survey of cyanobacterial blooms. For example, probes equipped with chlorophyll-a/phycocyanin and other sensors that can be coupled with autonomous monitoring platforms are now available for punctual or continuous estimations of cyanobacteria biomasses in the water column [6, 7]. In the same way, the use of remote sensing [8] and LIDAR [9] allows to estimate the horizontal and/or vertical distribution of the cyanobacterial biomass in a lake. Some of these tools remain under development but are increasingly used for cyanobacteria monitoring, particularly in developed countries. However, the cost of these devices is still a strong limitation to their use in developing countries [10].

Another strategy recently tested for the monitoring of water quality in freshwater ecosystems is the development of citizen science projects based on the involvement of the population living around the targeted freshwater ecosystems [11]. Some of these projects are specifically concerned with the monitoring of cyanobacterial blooms as they are easy to report by nonscientist actors due to changes occurring in the water color or the presence of colonies or aggregated filaments distinguishable by the naked eye (cf. Fig 1B). In the USA, the CyanoTRACKER project is based on the use of a smartphone application by volunteers to report algae blooms in their neighborhood pond and/or lake [12]. In the same way, citizen science programs based on the collection of water samples have been implemented for the monitoring of lake water quality and the survey of blooms in the province of Saskatchewan (Canada) [13] and in numerous lakes located in the state of Michigan (USA) [14].

Most of these projects take place in developed countries, fewer citizen programs being available in developing countries [15]. On the African continent, for example, a participatory local observatory has been implemented in Tanzania aiming at collecting and sharing data related to water, rainfall, fisheries, agriculture and food behavior between various stakeholders in water and land management [16]. In South Africa, Nare *et al.* [17] described the difficulties of community involvement and participation in water quality and management in a catchment area due to the inadequate institutional framework. Finally, a good relationship was found between the citizen monitoring of algal blooms and standard laboratory measurements in urban lakes and ponds of three cities located in Brazil and China [18]. These programs highlight the important benefits of this collaboration in terms of both knowledge production, opportunities for democratizing knowledge and expertise under the context of growing environmental and health risks [19].

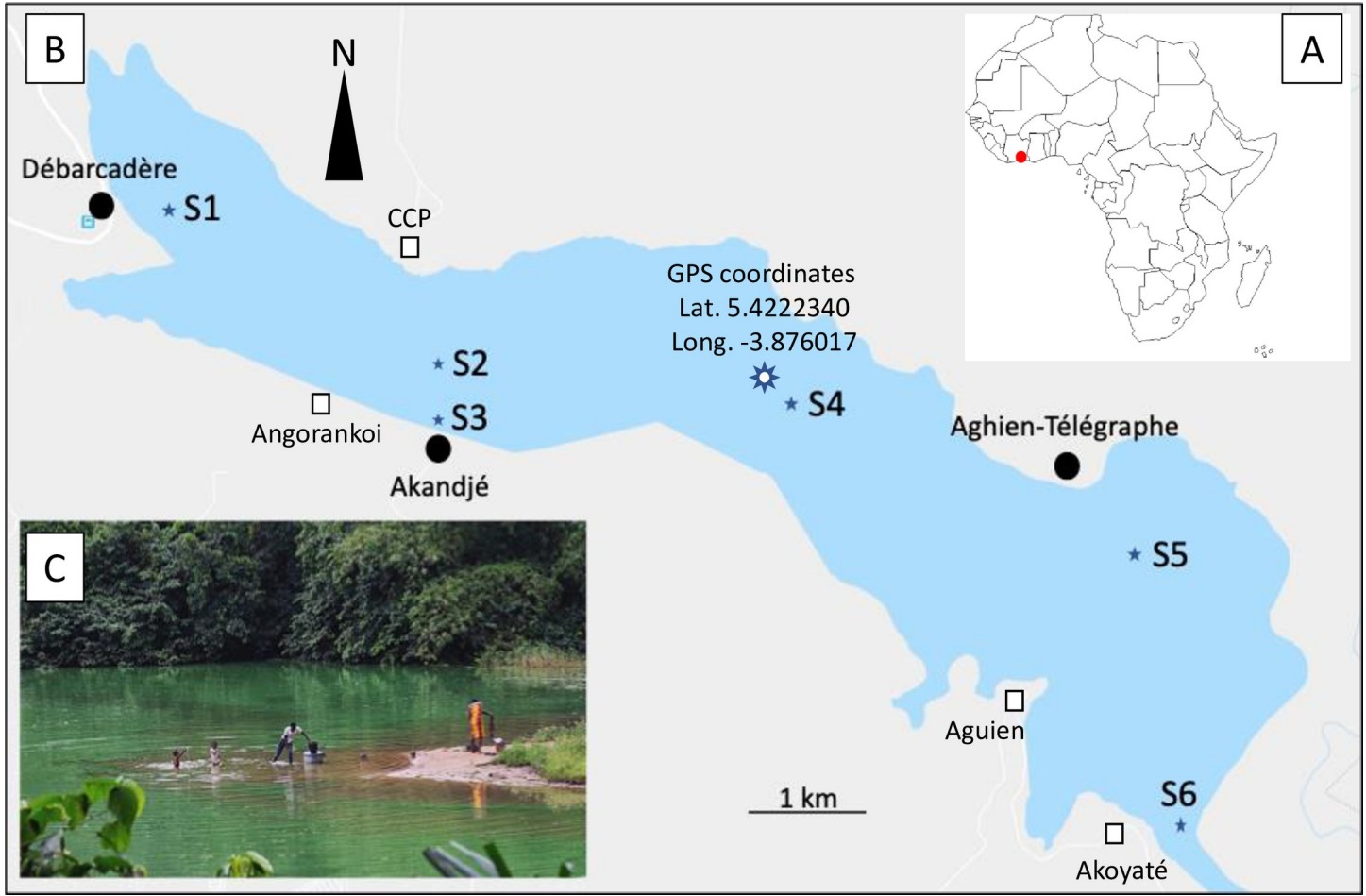

**Fig 1.** A. Location (red dot) of the lagoon Aghien in Ivory Coast. B. Map of the Lagoon Aghien with the locations of the three villages (Débarcadère, Akandjé, Aghien-Télégraphe) in which citizen monitoring of cyanobacteria was implemented (black circles: villages involved in the citizen monitoring; white squares: other villages located around the lagoon). C. Photo of a *Microcystis* sp. (cyanobacteria) bloom in a bay of the Lagoon Aghien.

In the framework of a collaborative project between three African countries (Ivory Coast, Senegal and Uganda) and France (WaSAf project, https://humbert19.wixsite.com/wasaf), we performed a first pilot study on the feasibility of setting-up a citizen monitoring of cyanobacteria in the African context. The three main goals of our study were to evaluate (i) the potentials and constraints for the mobilization of the local populations for the visual survey of cyanobacteria in aquatic ecosystems, (ii) the consistency between the data provided by the citizen monitoring of cyanobacteria and those obtained using a classical approach based on water sampling and cell counting and (iii) the contribution of citizen monitoring to awareness in the neighboring populations regarding the cause and health consequences of cyanobacterial blooms. This citizen monitoring approach was based on the use of a smartphone application and implemented in three villages located around a freshwater lagoon (Lagoon Aghien) in Ivory Coast.

## Material & methods

### Study site

Lagoon Aghien is a freshwater ecosystem located in the northeastern part of the city of Abidjan (Fig 1A). This lagoon covers a surface area of 19.5 km$^2$ with a perimeter of 40.72 km and a

volume of 25 km$^3$ [20]. Lagoon Aghien is part of a network of lagoons surrounding the city of Abidjan, with the particularity of being the only one containing freshwater. This lagoon is considered a potential water resource for the city of Abidjan in complement to the water supply currently provided by groundwater extraction. Approximately 12,000 people live in the immediate proximity of the lagoon, mostly distributed in eight villages located on its shoreline (Fig 1B). The lagoon is used by local populations for various domestic (bathing, washing, cooking, waste management, etc.) and economic (fishing, agriculture, etc.) practices that have an impact and are impacted by the water quality of the lagoon. Due to growing human populations and activities, not only in the immediate environment of the lagoon but also in its whole watershed, coupled with a dramatic lack of sanitation infrastructure and changes occurring in the land use occupation, Lagoon Aghien is now eutrophicated, and cyanobacterial blooms are frequently observed (Fig 1C). Despite the crucial importance of the lagoon for drinking water production, no institutional monitoring of water quality is currently performed.

The villages Débarcadère (3000 people), Akandjé (900 people) and Aghien-Télégraphe (1850 people) were chosen for the implementation of citizen monitoring because of (i) their geographical location enabling satisfactory coverage of the lagoon (Fig 1A), (ii) the local authorities of the three villages agreed to participate in this experiment and (iii) these villages were representative of the social organization and the various uses of the lagoon for all the villages.

Stars S1-S6 w performed by the Pasteur Institute of Abidjan

## Ethic statement

This study was not submitted for approval by an institutional review board (ethics committee) because there is currently no committee in Ivory Coast for environmental studies. But as described latter in the paper, we organized in each of the three villages concerned by this study, a meeting with the chiefdoms in order to present the project and to ask them for their agreement before to begin to work. In addition we asked also to the chiefdoms to designate the sentinels (referent people) in each village who have provided most of the data. Finally, our smartphone application entailed the possibility to collect anonymous reports.

## Social investigation for participative water quality monitoring

Sociological investigations conducted in 2017 within the WaSAf program have been used to prepare for the implementation of the participatory monitoring in the three villages. A sociological survey based on a semi-structured questionnaire was performed in all the villages located around the lagoon in order to better understand the different uses of the lagoon and the social organization of the local populations. For the purpose of this pilot study on the citizen monitoring of cyanobacteria, we only used the two questions dealing with the perception of the water quality by the local populations (See S1 Table).

According to the recommendations of Sturm et al. [21], various actions were performed with local populations with the goal to prepare, conduct and ensure the follow-up and finally report the results of the participatory monitoring in the three villages (Table 1; Fig 2). Firstly, meetings and semi-structured interviews were conducted with local community leaders and chiefdoms, and with inhabitants of the three villages in order to discuss about their interest for the implementation of a citizen monitoring of cyanobacteria. Secondly, focus groups were conducted with the more specific objective of identifying the opportunities and constraints for implementing participatory monitoring. These focus groups were organized with homogenous social categories according to gender and age (women, men and youths), including between 7 and 20 participants depending on the groups and the villages. They have permitted to find a

**Table 1. Description of the sociological investigations conducted during the implementation and the follow-up of the participative monitoring of cyanobacteria in the Lagoon Aghien.**

| | Débarcadère | Aghien-Télégraphe | Akandjé |
|---|---|---|---|
| **Questionnaires** | 46 | 44 | 29 |
| **Meetings with local chiefdoms** | 2 | 3 | 2 |
| **Focus groups** | 3 (women, men, youth) | 3 (women, men, youth) | 2 (women, men & youth) |
| **Training on the use of the smartphone application** | 1 | 1 | 1 |
| **Data restitution** | 2 | 2 | 2 |
| **Follow up interviews** | 6 | 7 | 3 |

common language between scientists and inhabitants, to better define the questionnaire in the mobile application and to generate enthusiasm among the participants to collaborate. Thirdly, a meeting was organized in each village with the goal to present and test the smartphone application with volunteers. Flyers explaining the use of the application were also distributed to participants to encourage a large participation.

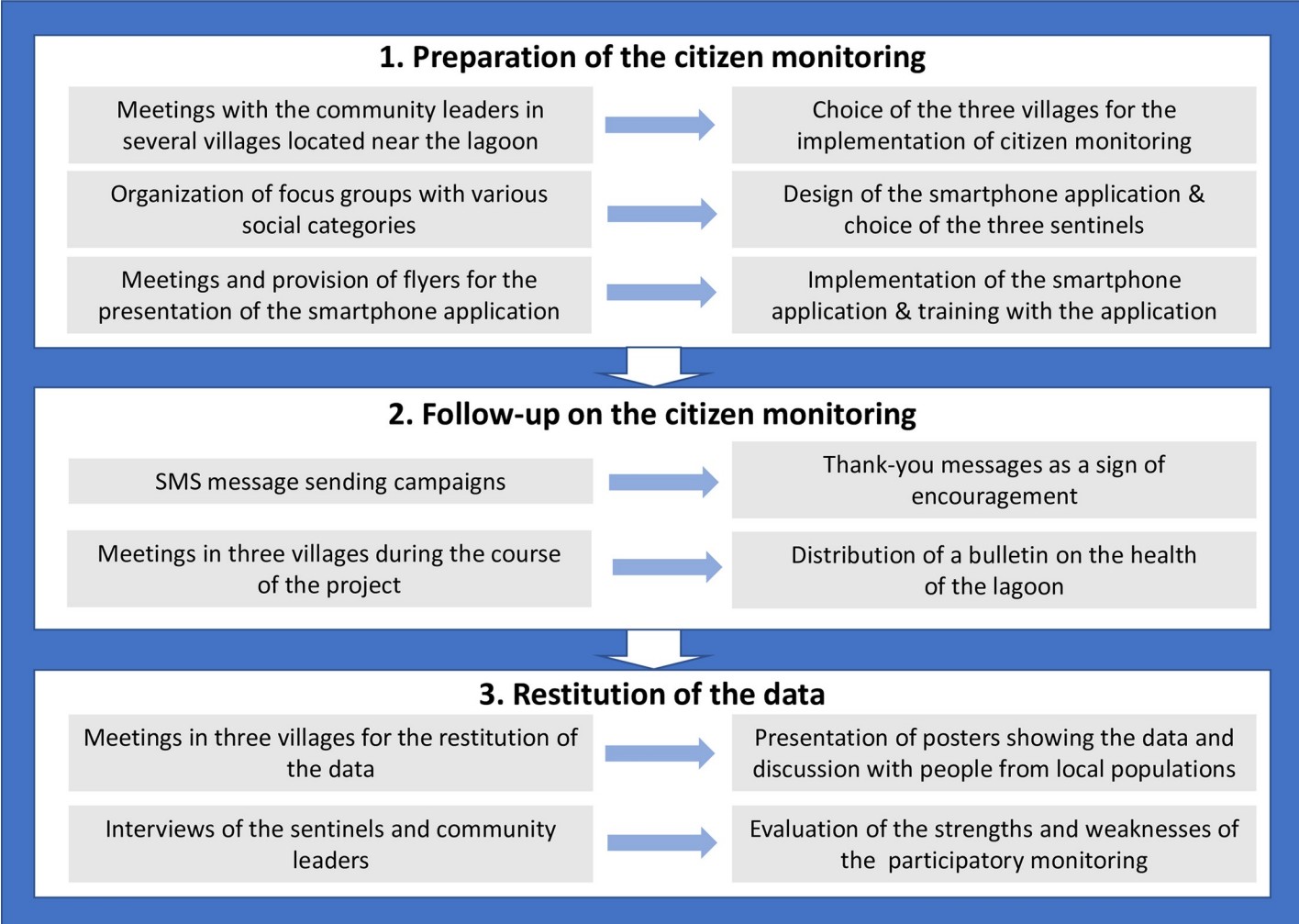

**Fig 2. Global overview of the different stages in our approach on the implementation, follow-up and data restitution of citizen monitoring of cyanobacteria in the three villages of the Lagoon Aghien.**

To ensure regular data on the water quality of the lagoon, one person, designated as a sentinel, was chosen in each village by the local chiefdoms. In Débarcadère village, the sentinel was a school teacher, while in Akandjé village, he was the representative of the young people. Finally, in Aghien-Télégraphe, the chief of the village decided that he could fulfill this task. Each sentinel was equipped with a mobile phone provided by the WaSAf program and was encouraged to take at least one picture per week of the lagoon, always in the same location (chosen together with the research team). The sentinel was also in charge of facilitating the participation of other members of the community willing to participate in the monitoring (helping them to install and use the application, explain the interest in the monitoring, etc.).

To maintain the interest of participants and encourage them to continue contributing, we sent SMS messages of thanks to the contributors having registered their phone number in the smartphone application twice during the experiment. In accordance with the project team commitments and to facilitate data appropriation in the villages, two restitution events (based on a poster presentation) were organized in each village at the end of each year of monitoring. We also prepared two times a prospectus (called Lagoon Health Bulletin) presenting together the data collected by citizens and those obtained from the water laboratory analyses performed by the Pasteur Institute of Abidjan. Finally, some educative actions explaining cyanobacterial blooms were performed in schools of Débarcadère and Aghien-Télégraphe villages.

At the end of the two years of the pilot study, we conducted qualitative interviews with the three sentinels and other volunteers who participated in monitoring to evaluate the effectiveness and difficulties related to participatory monitoring, as well as their appropriation of the monitoring experience. Our questions concerned (i) their participation in the monitoring (how they personally experimented the monitoring process and how this was concretely done, difficulties encountered, etc.), (ii) the impact of this participation on their own perception and understanding of the lagoon's ecological state and (iii) suggestions for improving the monitoring and its sustainability.

The participatory monitoring was organized in two periods: (i) during the first period (from October 2017 to August 2018), the sentinels were asked to provide at least one report per week, and (ii) during the second period (from September 2018 to August 2019), the sentinels were asked to collect and send reports only when something interesting concerning the water quality in general and/or the presence of a cyanobacterial bloom was detected. However, some participants other than sentinels have continued to send reports, even when there was no cyanobacterial bloom. We chose to change the strategy in the middle of the pilot study in order to determine if the sentinels remained vigilant about cyanobacteria when they no longer had the obligation to send a weekly report.

## Smartphone application

The generalized access to mobile phone services endowed with geographic information systems (GIS) creates new opportunities for data collection, access and sharing all over the world [22–24]. According to data from the World Bank [25], access to cellular networks has reached 90% in numerous African countries, while access to drinking water is only approximately 86% and access to sanitation is close to 60%. Consequently, the use of a smartphone application seemed to be a good tool for involving local communities in participatory monitoring. We adapted a free smartphone application that is part of the Epicollect5 data-gathering platform (https://five.epicollect.net/) to the specific needs of the monitoring. This solution allowed the generation of project-dedicated questionnaires and the collection of georeferenced data that could be displayed through a web application in the form of maps, tables and charts. For each entry in the WaSAf project (S1 Fig), users completed a short questionnaire asking for the

presence of (i) a green water color and a visible mass of algae suspended in the water, (ii) traces of fuel contamination, (iii) dead fish, (iv) garbage accumulation and/or if there was nothing to report or if they wanted to add other observations. For each questionnaire, users were asked to update their geo-location and to add a picture if they wished. Users can choose to respond anonymously but they could also provide their phone number. This optional information could be used by the project team to provide feedback and to confirm the importance of the collected data (two SMS campaigns were sent). At the end of each questionnaire, users recorded their data and sent them to the website if a data connection was available. If there was no internet connection, the recorded data could be sent later. As soon as a questionnaire was sent, the data were available online. All the data collected in the framework of this project are available at the project website: https://five.epicollect.net/project/wasaf.

### Characteristics of the monitoring of cyanobacteria performed by the Pasteur Institute of Abidjan

In the framework of the WaSAf project, a monitoring of the water quality in the Lagoon Aghien was performed at six sampling stations (S1-S6) from December 2016 to April 2018 by the Pasteur Institute of Abidjan (Fig 1A). Consequently, it was only possible to compare from October 2017 to April 2018, the phytoplankton data obtained by this monitoring with those resulting from the citizen monitoring of cyanobacteria. Data collected on the phytoplankton community were based on an integrated water sampling performed in the first meter of the water column according to the technique used by Laplace-Treyture *et al.* [26]. The counting of cyanobacteria was performed on an inverted microscope (Nikon Eclipse TS100) using the Utermöhl method, according to the AFNOR 15204 standard. A minimum of 30 randomly selected fields was counted to attain at least 500 counting units (cells, colonies or filaments/trichomes).

The results were then expressed as the number of cells per milliliter. In the case of the colonial cyanobacteria *Microcystis* sp., two categories of colonies were considered regarding their diameter (< or > 200 μm). The mean number of cells per category was estimated under an upright microscope by counting the number of cells in 30 colonies per category after they were gently crushed between slides and coverslips. For the filamentous cyanobacteria *Aphanizomenon*, *Cylindrospermopsis*, *Limnothrix*, *Lyngbya*, *Oscillatoria*, *Planktothrix*, *Pseudanabaena* and *Dolichospermum*, the counting considered filament lengths of 100 μm. The mean number of cells per 100 μm of filament was estimated under an upright microscope for 30 filaments for each genus.

The mean cell biovolumes were calculated for the cyanobacteria genera that were most abundant in Lagoon Aghien (*Microcystis*, *Cylindrospermopsis*, *Limnothrix*, *Oscillatoria*, *Dolichospermum*) from measurements performed on 30 cells per genus. For each of the other genera, we used standard cell values defined in the HELCOM PEG Biovolume reports (http://ices.dk/marine-data/vocabularies/Pages/default.aspx).

## Results

### Overall presentation of the evolution of the number and locations of collected reports during the study period

During the first period of this pilot study (from October 2017 to August 2018), 443 reports have been recorded in the WaSAf project hosted by the Epicollect5 platform (see https://five.epicollect.net/project/wasaf). Among these 443 reports, phone numbers were only provided in 37 of them. During the second period of the study (from September 2018 to August 2019), 171

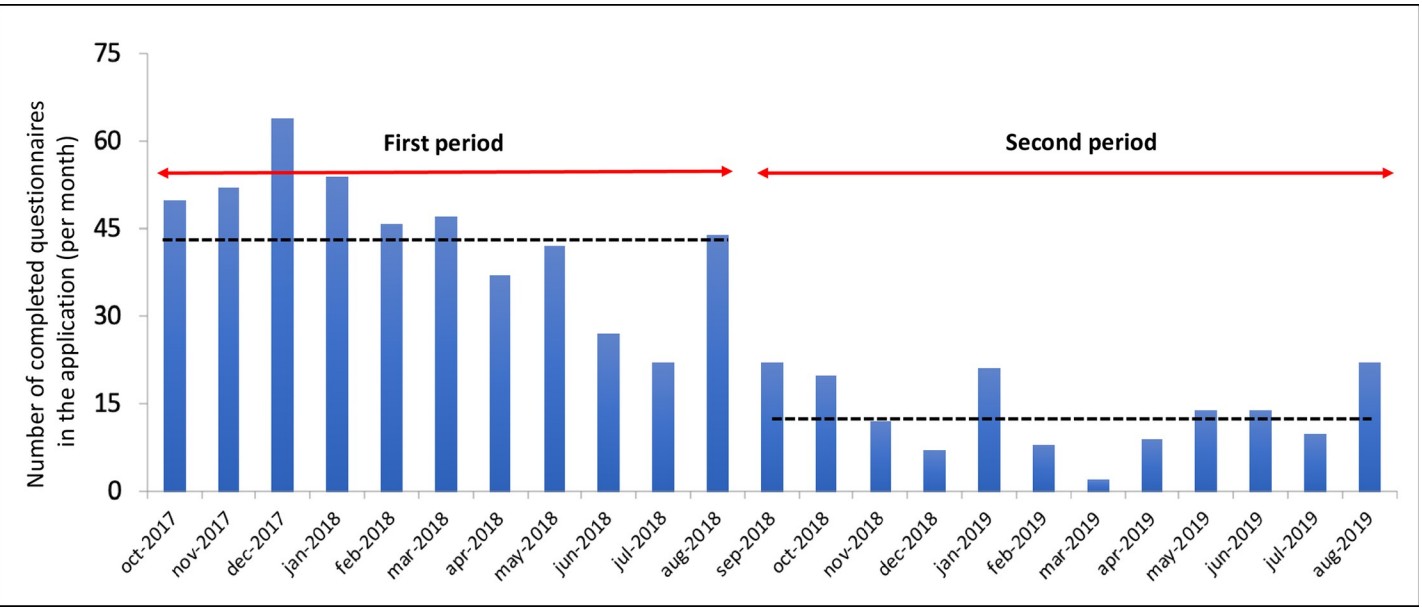

**Fig 3. Number of completed questionnaires loaded on the WaSAf project Epicollect5 platform.** During the first period, the sentinels were asked to complete at least one questionnaire per week with one picture of the lagoon. During the second period, sentinels were advised to send pictures only if cyanobacteria or other worrying phenomena were observed. Dashed line: Monthly average of collected questionnaires during each period.

questionnaires were collected and phone numbers were provided in only 7 of them. During the whole study period, it appears that 92.8% of the collected data with the smartphone application were anonymous, which makes it impossible to estimate the number of participants who have contributed to the monitoring.

During the first period of the study, an average of 44 reports was collected each month in the web site of our application, while during the second period, an average of 13 reports were collected monthly (Fig 3). This difference in the number of collected reports during the two periods was expected since we changed the recommendation for the sentinels from sending a picture at least once a week (regardless of whether cyanobacteria were observed) to sending pictures only if cyanobacteria or other degradation signs were observed.

As shown in Fig 4A, the geographical origins of the data collected during this study were distributed all around the lagoon and not only near the three villages. Many data points were collected in the western part of the lagoon by the sentinel from the Débarcadère village (Fig 4B). This sentinel performed visual monitoring at four points in this part of the lagoon. Finally, it also appears on the map that some of the data were not located on the shoreline of the lagoon.

## Data indicating the potential presence of cyanobacterial blooms

The potential presence of cyanobacterial blooms in the lagoon was reported in the smartphone application by the response "Green water, visible algae" in the questionnaire. These responses were sometimes supported by the pictures associated with the questionnaire (see for example two pictures in S2 Fig) but frequently the poor quality of pictures did not permit us to confirm the presence of cyanobacteria. As shown in Fig 5, cyanobacteria blooms were reported mainly during four periods. The first peak of reports occurred in November 2017, followed by a small peak in February 2018, a third peak in May-June 2018 and finally a fourth large peak from May to September 2019.

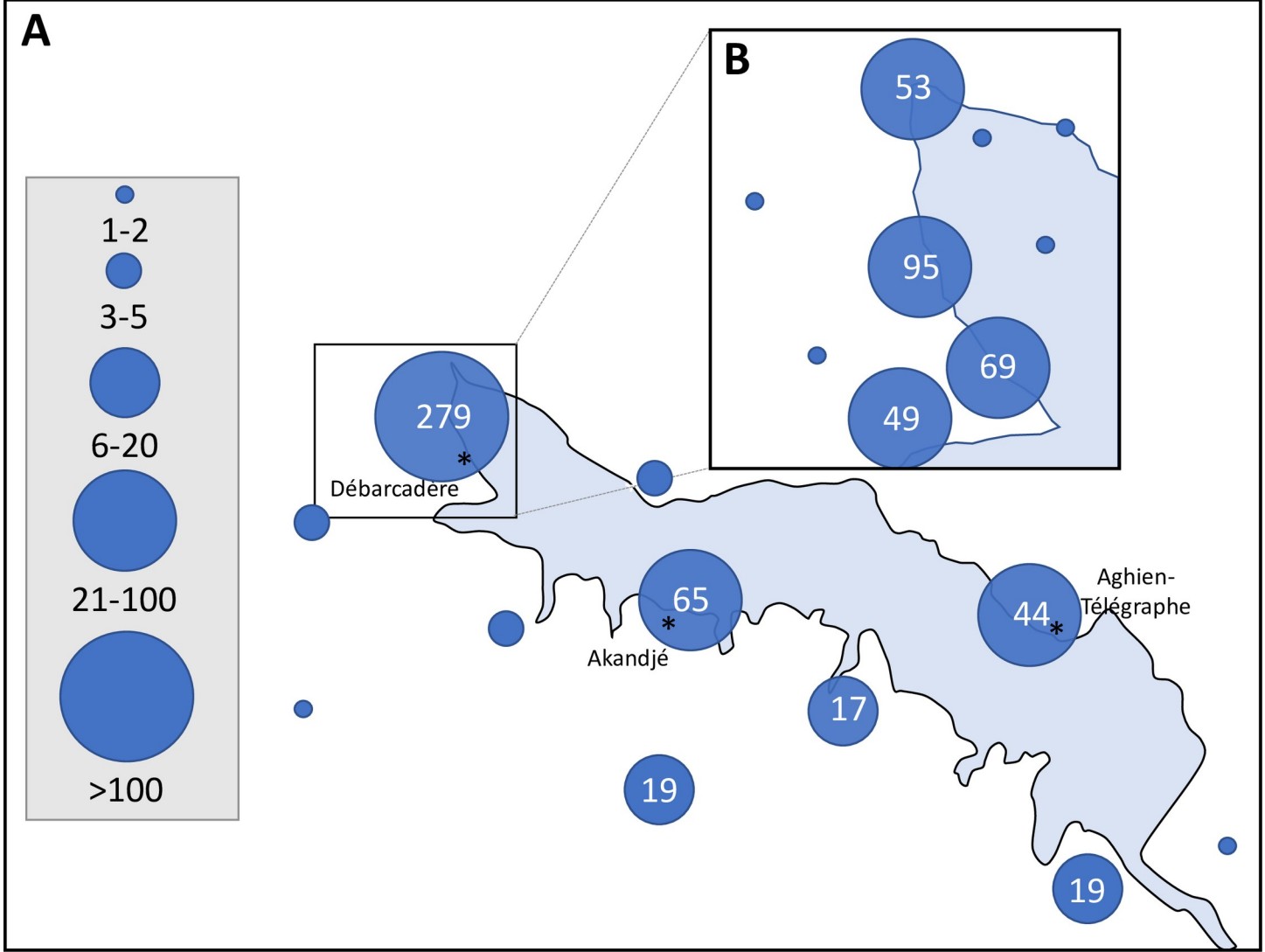

**Fig 4.** Distribution of the reports collected by citizens around the Lagoon Aghien (A). Detail on the citizen monitoring performed by the sentinel from Débarcadère village (B). The size of the circles is proportional to the number of reports collected in each place. For the places where more than five reports were collected, the exact number of collected reports is provided into the circle.

The geographical locations of the potential algal blooms indicated by the presence of a green water color/visible algae during the three periods were not the same. For the first peak (November 2017), almost all reports were from the Débarcadère village, while for the two other peaks (May-June 2018 and May-September 2019), almost all reports of algal blooms were from the Aghien-Télégraphe village.

## Comparison of the data provided by the participatory monitoring with those from the Pasteur Institute

It was difficult to compare the data collected by the Pasteur Institute of Abidjan with that collected by the citizens. Indeed, the citizen monitoring was performed on the shores of the lagoon while the monitoring of cyanobacteria by the Pasteur Institute of Abidjan was performed in six sampling points located more than 150 m away from the shores (except for S3). Moreover, water

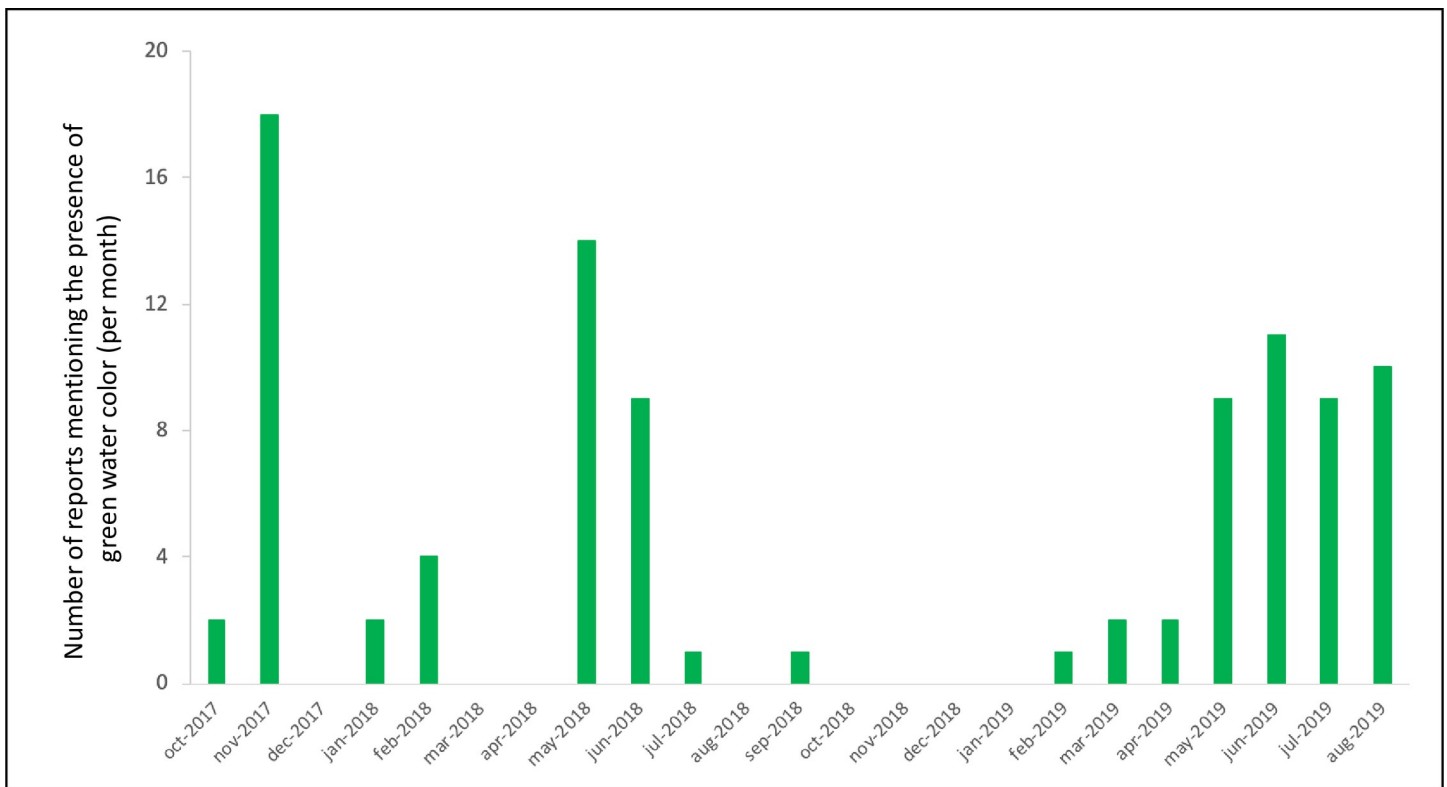

**Fig 5. Evolution of the number of reports made by citizens on the potential presence of cyanobacteria in the Lagoon Aghien (green water color and/or presence of visible colonies).**

samples were collected by Pasteur Institute once a month while the smartphone reports were sent by citizens as the days went by. For these reasons, we have only compared the monthly number of reports mentioning the presence of a green water color/visible algae with the monthly variations in the total cyanobacterial biovolumes estimated at each sampling stations.

Two increases in cyanobacterial biovolumes were observed during the monitoring performed by the Pasteur Institute from October 2017 to April 2018. The first increase was recorded in November 2017, and the second one was recorded in February 2018 (Fig 6). Interestingly, during these two increases of cyanobacterial biovolumes, there was also an increase in the number of citizen reports mentioning green water color/visible algae. During the seven months of the Pasteur Institute monitoring, cyanobacteria dominated the phytoplankton community, representing on average 40% (±18) of the total phytoplankton biovolumes estimated at each date and sampling station, when Dinophyta, Chlorophyta, Bacillariophyta and Euglenozoa represented respectively 19% (±14), 17% (±11), 15% (±17) and 9% (±6) of these biovolumes. Finally, the data collected by Pasteur also highlighted that large variations occurred in cyanobacterial biovolumes depending on the sampling stations, with the largest biovolumes being recorded during this study period at Station 4 and the lowest at Station 6.

## Local understanding of the issues of water quality and cyanobacteria in the lagoon

From the social investigation performed before the implementation of the citizen monitoring, significant differences (Chi2 test, p<0.0001) were found in the opinions about the water

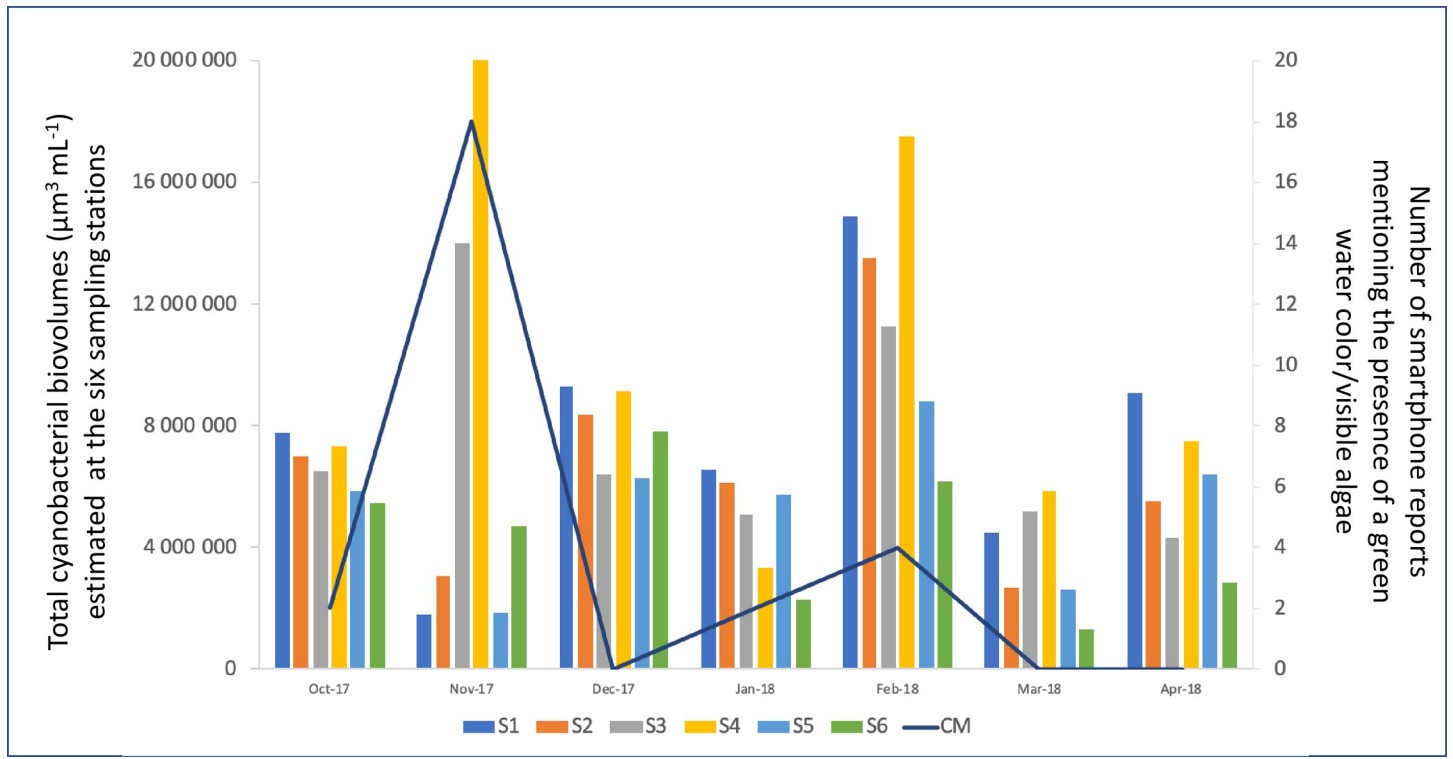

**Fig 6. Cyanobacterial biovolumes estimated at the six sampling stations of the Lagoon Aghien by the Pasteur Institute and number of reports mentioning a green water color / visible algae made on the smartphone application by citizens.** CM = Citizen monitoring. S1 to S6 = Sampling stations 1 to 6 (see Fig 1).

quality of the lagoon in the three villages (Fig 7A). Almost 90% of the interviewed people from the Akandjé village and 56% from Aghien-Télégraphe told us that the lagoon exhibits "good and very good quality". On the other hand, the quality perception was more negative in the Débarcadère village (72% of interviewed people think that the lagoon is in a bad or very bad state).

Significant differences were also found between the three villages in the criteria used by local populations to qualify the recent evolution of the water quality in the lagoon (Chi2 test, p<0.001). These differences mainly concern the proliferation of reeds and the bad smells, which were more specifically associated with the degradation of the water quality in Aghien-Telegraphe and Débarcadère respectively (Fig 7B). On the other hand, the two most cited criteria used to qualify the recent degradation of the water quality in the three villages were the fishing captures (a decrease over the last 20 years according to their perception) and the changes occurring in the water color (Fig 7B). The relative importance of these two criteria varied according to the villages, with significant differences between Akandjé and the two other villages (Chi2 test, p<0.05).

Finally, the focus groups and qualitative interviews have revealed that two main changes in the water color are clearly identified by the local populations: (i) brown water (muddy, reddish) observed mainly during the rainy seasons (from May to July and from September to November), and (ii) green water observed before and/or after the rainy seasons. The brown color was (mainly) associated with "dirty water" coming from the watershed during the rainy periods. During these periods, inhabitants take precautions, and the lagoon is less used for domestic activities to avoid water-related diseases such as diarrhea. The green color of the lagoon was not associated with any sanitary or environmental risks. In contrast, some

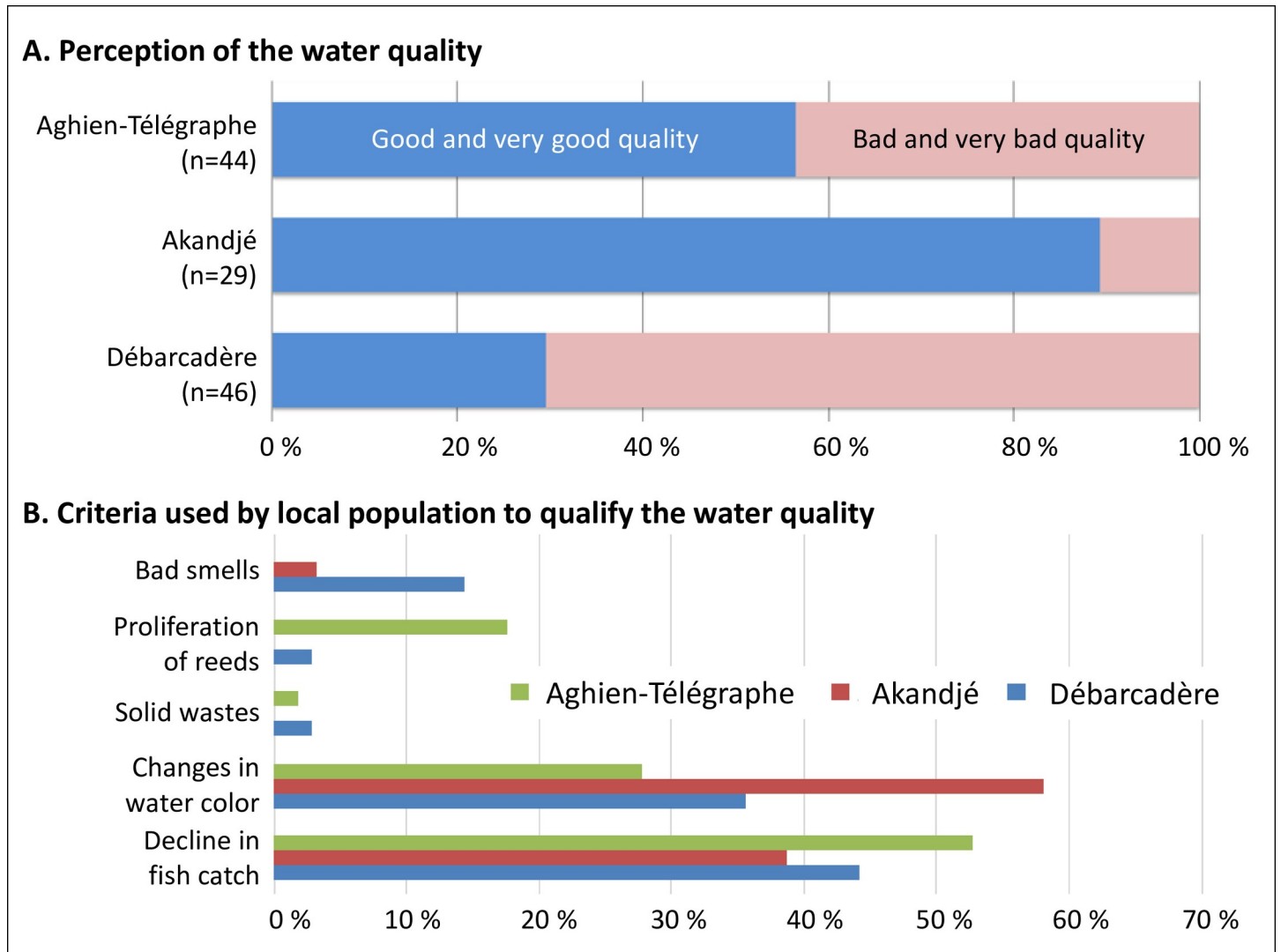

**Fig 7. Perception of the water quality in Lagoon Aghien by residents of the three villages (Aghien-Télégraphe, Akandjé, Débarcadère).** Respondents were asked (A) how they evaluate the quality of the lagoon using a four-point scale (very good, good, bad, very bad) and (B) which criteria they use for the assessment of the water quality.

participants noticed a coincidence between green water color and fish abundance: "*When the water turns green, it smells such as fish, it is fish poop, it is to tell the fishermen that there is a lot of fish and that they can go out fishing*" (Focus group women, Akandjé, Nov. 2017). Only some of the villagers noticed that bathing activities during these green water periods could generate skin and eye irritation. Consequently, no or very few precautions related to domestic uses of water are taken during cyanobacteria blooms. Finally, many participants of the focus groups made a link between the brown color (the runoff pollution) and the green color (eutrophication), with the first color often preceding the second.

The initial focus groups and the restitution events represented important opportunities for explaining the reasons why researchers and members of the Pasteur Institute were so interested in the monitoring of the "green spots", as local people called them. On the one hand, local observations are relevant (the phytoplankton biomass production fosters the fish populations), but it has to be stated and explained that a massive development of cyanobacteria also

represents a sanitary risk for human activities and can be considered a good indicator of the overall evolution of the water quality in the lagoon. On the second hand, such moments of exchange are necessary to motivate the local communities and raise local concern about cyanobacterial blooms, which were previously considered as a natural phenomenon.

Finally, the initial focus groups and the restitution events played a role of "hybrid arenas" for the communities and scientists to discuss about the causes and consequences of the degradation of the water quality in the lagoon with scientists. Local chiefdoms were very willing to discuss about the contribution to the pollution of the lagoon by populations living close to the lagoon compared to that originating from the rest of the watershed and the actions and programs that could be implemented to protect/restore the water quality in the lagoon.

## Discussion

Although the participation of nonscientists in environmental monitoring is not a recent practice, the phenomenon has undergone spectacular development over the last twenty years, facilitated by new communication technologies and mobile computing applications. This first citizen project based on the use of a smartphone application for the monitoring of cyanobacterial blooms in Africa shows that there is an opportunity for such approaches for both (i) the improvement of the monitoring of cyanobacteria in addition to the monitoring performed by relevant institutions and (ii) the improvement of the awareness and social learnings for the local communities concerning the health risks associated with cyanobacteria and more globally about the degradation of the water quality.

Concerning the improvement of institutional monitoring, as mentioned in the introduction, it is well known that accurate monitoring of cyanobacterial blooms is very costly. This financial cost is a major constraint for the implementation of long-term monitoring in developing countries because of the difficulty of securing long-term funding in these countries. The best illustration of this issue was that the monthly monitoring performed by the Pasteur Institute of Abidjan, at the same time as the participatory monitoring implementation, was stopped because of the lack of financial resources to continue it. Our pilot study has shown that it is possible to mobilize local populations living close to a freshwater ecosystem to monitor it. In this context, it is interesting to raise the issue of the place of this participatory monitoring in an overall monitoring system for the Lagoon Aghien. First, citizen monitoring could be an early warning system for cyanobacterial blooms, which could trigger water samples and analyses. This system could significantly (i) improve the efficiency of institutional monitoring based, for example, on a monthly sampling frequency in a limited number of sampling points and (ii) reduce the cost of institutional monitoring if these two approaches are well coordinated. Second, citizen monitoring coupled with education and capacitation programs could also permit improved basic knowledge of the dynamics and toxicity of cyanobacterial blooms and, more globally, the knowledge of water quality, when coupled with water sampling by untrained and equipped volunteers, as has been done in other participatory projects [13].

Concerning the local awareness and social learnings about cyanobacteria, the focus groups showed that local populations living around the Lagoon Aghien could spontaneously identify visible pollution, as well as changes in physical and biological parameters (such as the water color, macrophytes and algal blooms). At the same time, these focus groups also revealed the low awareness of these populations concerning the health risks associated with cyanobacterial blooms. During the preparation and feedback meetings organized in the three villages, there were also many discussions on the causes of cyanobacterial blooms in freshwater ecosystems. The issue of eutrophication was well understood by most of the participants, with interesting debates on the relative contribution of nutrient discharge into the lagoon from local pollution

*versus* "distant pollution" from the other parts of the Abidjan district located in the watershed. These discussions, for example, led the chiefdom representatives of one village to prohibit defecation in the lagoon by the inhabitants of this village and asked for sanitation investments from the regional authorities. Finally, it appears that the social benefits associated with the citizen monitoring approach are both direct (new learning and understanding of water ecosystem functioning and the eutrophication issue and its consequences to cyanobacterial blooms; new capacities to identify pollution sources and sanitary and ecological risks, etc.) and indirect (their appropriation and translation into conservation actions, etc.).

Several limits were also identified during this first pilot study on the citizen monitoring of cyanobacteria in the Ivory Coast. These limits can be qualified as (i) technical limits; (ii) participatory limits; (iii) data accuracy and validation limits and (iv) institutional limits.

The first technical limits are related to the accuracy of the produced data. Some data were affected by the approximate geolocations of several reports, which were not located on the shoreline of the lagoon. These geo-localization errors were probably due to the misuse or poor quality of the GPS of some smartphones, knowing that the rapid enhancement of smartphone quality will probably reduce this problem in the coming years. In the same way, the poor quality of some of the pictures associated with the questionnaires did not permit visual confirmation of the presence of "green algae" indicated in the questionnaire. Again, it is likely that the rapid enhancement of camera quality in smartphones [27] will permit us to solve this problem in the future. Finally, we observed during the meetings that most of the young men and women (<45 years old) had smartphones equipped with a camera, but this was not the case for people >45 years old, who had only small cellphone without a camera. It is likely that the decrease in the prices of smartphones will result in the future, in a more widespread availability of these devices in the populations.

The participatory limit in our pilot study concerns the fact that despite a very important number of registered data, most of them were supposed to be sent by the local sentinels. It was seldom possible to know who sent the reports because few people have provided their phone number in the reports. However, when the phone number was provided, it appeared that the sentinels of the three villages sent most of the reports. It is likely that some social/gender categories do not seem to contribute to citizen monitoring such as for example, women, although they are in close contact with the lagoon as they have multiple uses for it. However, from the interviews with the sentinels, it appeared that some villagers who observed changes in the water color asked the sentinels to create a report on the application. Thus, the role of sentinels was very important in the villages, which made the selection of sentinels very important. In our experiment, the best contributor as a sentinel was clearly the schoolteacher from the Débarcadère village, who self-planned to conduct multipoint monitoring. As noted by Requier *et al*. [28], the issue of long-term participation and the improvement of volunteer participation is facilitated by key actors (such as sentinels) who can act as intermediaries between scientists and local populations.

We identified several limits in our process of validating data provided by the citizen monitoring. Some congruence was found between these data and that collected by the Pasteur Institute but this comparison was not fully satisfactory. It would be preferable to perform a monitoring of the water in the same areas where green water color/visible algae were reported by citizens. To achieve this goal, we recommend two different methods for the validation of data collected by citizens. The first one will be based on the use of a smartphone spectrometer by the sentinels in order to evaluate some water quality parameters [29] while the second will consist of water collection and fixation with lugol's iodine solution, then a microscope examination by scientists. These two methods will be tested in France during the summer 2020 in

the framework of a citizen monitoring of cyanobacteria with the same approach used in Ivory Coast.

The institutional limit in our citizen monitoring approach concerns the integration of the data collected by the citizens within the framework of the institutional monitoring of the lagoon, which is actually missing. The two ministries in charge of the management of water resources and drinking water production have expressed great interest in this participatory approach but failed to see how the collected data could be integrated by their institutions. This result was not because participatory data were considered useless or not accurate enough but because the institutions do not devote the financial and human resources needed for long-term monitoring of freshwater resources. A great deal of awareness-raising among these institutions still needs to be done to show how citizen empowerment might be integrated in the framework of the implementation of long-term monitoring of freshwater resources. In this goal, we are currently working with the different water institutions of Ivory Coast (Water and Forests Ministry and Hydrology Ministry) for the implementation of an environmental observatory of the Lagoon Aghien, in which citizen monitoring would take its place as an early warning system for cyanobacterial blooms. We hope to reach this objective before the end of the year knowing that citizens have shown their willingness to continue the monitoring of cyanobacteria as shown by the fact that they still send reports while citizen monitoring ended officially one year ago.

Finally, in addition to the design of a smartphone application and the collection of data on cyanobacterial blooms in an African freshwater ecosystem, our citizen monitoring project considered the social aspects of participatory science. These social factors are crucial because they are both a condition for effective participatory monitoring and a condition for maximizing the social utility of the procedure. From this perspective, participatory science is much more than recruitment of volunteers for data collection. The empowerment of knowledgeable citizens, the adoption of critical thinking skills related to the environment and the strengthening of science-society relationship are acknowledged as one crucial point of participatory science projects, even if these points are sometimes marginal in the design of participatory science projects [30, 31].

In conclusion, this pilot study conducted in Ivory Coast has shown that it is feasible to mobilize local populations in an African country for the monitoring of cyanobacteria. In particular, the choice of having one referent sentinel per village seems to be a good strategy. We also show that this kind of approach, from its initial preparation with local populations to its realization, has improved their awareness on the global issue of water quality. One important issue arising from this study concerns the validation of the data provided by the citizens, which was not fully achieved. In order to solve this problem, we proposed to test two strategies for the validation of data, which will require collaboration between sentinels and scientists. Finally, the greatest challenge of the citizen monitoring of cyanobacteria but also more generally of all the participatory approaches, concerns their sustainability, knowing that this issue will be handled differently in developed and developing countries. In numerous developed countries where there is already an institutional monitoring of cyanobacteria, the challenge of the sustainability of the citizen monitoring is to show how it can contribute to improve the institutional monitoring of cyanobacteria in order to legitimize its place. For most of the developing countries where there is no existing institutional monitoring, the challenge is to coordinate the joint implementation of these two approaches (institutional and citizen) in order to maximize their efficiency and to minimize the global cost of the monitoring, which is a key point for its sustainability.

## Supporting information

**S1 Fig. Screenshots of the Epicollect-5 smartphone application with the WaSAf project.**
(PPTX)

**S2 Fig. Two examples of bloom pictures taken by the participants during the citizen monitoring of the Lagoon Aghien (all the pictures are available on the website (https://five.epicollect.net/project/wasaf).**
(PPTX)

**S1 Table.**
(DOCX)

## Acknowledgments

We are extremely grateful to the three sentinels (Mr. Blaise Lattho in Aghien-Télégraphe, Mr. Kouandjo Guy Rolland Adon in Akandjé, and Mr. Kameleman Siaka Issoufou in Débarcadère) and the inhabitants and chiefdoms of the three villages (Aghien-Télégraphe, Akandjé and Débarcadère) for their decisive contribution to this work. Finally, we would like also to thank the three reviewers of this paper for their constructive comments.

## Author Contributions

**Conceptualization:** Veronica Mitroi, Pierre-Yves Bulot, José-Frédéric Deroubaix, Julien Coulibaly Kalpy, Jean-François Humbert.

**Data curation:** Veronica Mitroi, Pierre-Yves Bulot, Mathias Koffi Ahoutou, Catherine Quiblier, Jean-François Humbert.

**Formal analysis:** Veronica Mitroi, Kouadio Chrislain Ahi, Pierre-Yves Bulot, Fulbert Tra, José-Frédéric Deroubaix, Mathias Koffi Ahoutou, Catherine Quiblier, Jean-François Humbert.

**Funding acquisition:** Jean-François Humbert.

**Investigation:** Veronica Mitroi, Kouadio Chrislain Ahi, Pierre-Yves Bulot, Fulbert Tra, José-Frédéric Deroubaix, Mathias Koffi Ahoutou, Jean-François Humbert.

**Methodology:** Veronica Mitroi, Pierre-Yves Bulot, Fulbert Tra, José-Frédéric Deroubaix, Jean-François Humbert.

**Project administration:** Mariatou Koné, Julien Coulibaly Kalpy, Jean-François Humbert.

**Resources:** Kouadio Chrislain Ahi, Pierre-Yves Bulot.

**Software:** Pierre-Yves Bulot.

**Supervision:** Veronica Mitroi, Fulbert Tra, José-Frédéric Deroubaix, Catherine Quiblier, Mariatou Koné, Julien Coulibaly Kalpy, Jean-François Humbert.

**Validation:** Veronica Mitroi, Kouadio Chrislain Ahi, Pierre-Yves Bulot, Fulbert Tra, José-Frédéric Deroubaix, Mathias Koffi Ahoutou, Catherine Quiblier, Jean-François Humbert.

**Visualization:** Veronica Mitroi, Catherine Quiblier.

**Writing – original draft:** Veronica Mitroi, Jean-François Humbert.

**Writing – review & editing:** Veronica Mitroi, Kouadio Chrislain Ahi, Pierre-Yves Bulot, Fulbert Tra, José-Frédéric Deroubaix, Mathias Koffi Ahoutou, Catherine Quiblier, Mariatou Koné, Julien Coulibaly Kalpy, Jean-François Humbert.

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
