## [Decision Letter · Decision Letter 0]

26 May 2020

PONE-D-20-09032

Can participatory approaches strengthen the monitoring of cyanobacterial blooms in developing countries? Results from a pilot study conducted in the Lagoon Aghien (Ivory Coast)

PLOS ONE

Dear Dr. Humbert,

Thank you for submitting your manuscript to PLOS ONE. After careful consideration, we feel that it has merit but does not fully meet PLOS ONE’s publication criteria as it currently stands. Therefore, we invite you to submit a revised version of the manuscript that addresses the points raised during the review process.

We look forward to receiving your revised manuscript.

Kind regards,

Cleber Cunha Figueredo, Ph.D.

Academic Editor

PLOS ONE

Journal Requirements:

1, Please ensure that your manuscript meets PLOS ONE's style requirements, including those for file naming. The PLOS ONE style templates can be found at

2. We note that Figure S1 includes an image of a participant in the study. 

Additional Editor Comments (if provided):

This is an interesting study focused on citizen science approach and based to the observations made by populations from three villages in Ivory Coast. However, it is necessary more information to guarantee that the method is really efficient to detect cyanobacteria blooms with no classic monitoring tools. Some tools should be used to make clear the comparisons between visual observations and algae quantification. Authors should give more information on the algal community structure.

It is necessary to do major corrections to improve the MS and I suggest the authors consider all suggestions made by reviewers. It is also important to do a point by point correction to inform all changes in the text and to answer all questions pointed by reviewers.

Reviewers' comments:

Reviewer's Responses to Questions

**Comments to the Author**

1. Is the manuscript technically sound, and do the data support the conclusions?

Reviewer #1: Partly

Reviewer #2: Yes

Reviewer #3: Partly

2. Has the statistical analysis been performed appropriately and rigorously? 

Reviewer #1: No

Reviewer #2: N/A

Reviewer #3: No

3. Have the authors made all data underlying the findings in their manuscript fully available?

Reviewer #1: Yes

Reviewer #2: Yes

Reviewer #3: Yes

4. Is the manuscript presented in an intelligible fashion and written in standard English?

Reviewer #1: Yes

Reviewer #2: Yes

Reviewer #3: Yes

5. Review Comments to the Author

Reviewer #1: The manuscript submitted to be published as a research article in PlosONE by Humbert et al. presents results on participatory citizen monitoring of algal blooms in the Lagoon Aghien in the Ivory Coast, Africa. The work is done as part of a larger WaSAf project in the area. The objectives of the study were to evaluate the potentials and constraints for the mobilization of the local population around the Lagoon Aghien for the visual monitoring of cyanobacteria/algal occurrence, to compare the results of the cyanobacteria occurrence obtained by the citizen monitoring against the results of phytoplankton sampling and analysis done by the Pasteur Institute of Abidjan (a 7 month overlapping period), and to evaluate the commitment and experiences of local people on the citizen monitoring, and how the process and participation influenced their awareness on cyanobacterial blooms and risks.

The subject of the paper is highly topical as potentially toxic cyanobacterial blooms are a big risk for the use of water in particular in the less developed countries. Therefore, the means to monitor cost-efficiently their occurrence and risks as well as to increase awareness among citizens, water users and water operators is crucial.

The paper is clearly written and easy to follow. The importance of social factors in implementation of this kind of citizen monitoring is well emphasized. Unfortunately, at present the paper is purely descriptive. The authors should use statistical tools to analyze the comparability/congruence between the visual observations and the cyanobacteria biomass results from the selected stations. I appreciate the manuscript is more on describing the process itself, i.e. setting of a participatory monitoring network in a less developed area and to increase awareness in an undeveloped region, but as algal observations and their comparison form an integral part of the paper they should be properly analyzed and presented (see also my later comment to Fig. 6). This comparison could be reasonable to do by analyzing the visual observations from Débarcadère against the station S1 data, from Akandjé against the stations S2 and S3 data, and from Aghien-Télégraphe against the stations S4 and S5 data. This should more reliable show the level of congruence.

There are similar “quantitative” weaknesses related to the participatory (“socio”) part of the work. For instance, the manuscript lacks detailed statistics; how many people were involved altogether for the monitoring, and what was the number of people who attended the awareness rising gatherings.

In the study, algal bloom situation (cyanobacteria mass occurrence) was monitored as based on the water color (changes). For the reader it would be important to know, what was the contribution of cyanobacteria of the total phytoplankton biomass during the 7 mo period when the biomass data was available for comparison. This information should be given. Further, in lakes algal blooms and water coloration can also result from mass occurrences of other algal groups like chlorophytes, diatoms, euglenoids and occasionally even cryptophytes. How this can be taken into consideration in the risk assessment and interpretation of smartphone photos. This issue needs to be discussed more in the paper. Now it is only briefly referred on page 23.

The essential Figure 6 and its legend should be improved and redrawn. The phytoplankton biomass results from different stations (S1-S6) are overlapping (or are they stacked?). For instance, the bloom situation on Nov-17 was observed in the Débarcadère area, where the nearest sampling station for cyanobacteria biomass monitoring was S1. However, in Fig. 6 the biomass from the station S4 hides completely the results of station S1. Also, it should be clearly said in the legend what the citizen observation (CM) bars represent: a monthly average of all observations from the three villages, or something else. Also, it might be more informative to present all data as bars, and including also standard deviation and n of observations? The primary y-axis of Fig. 6 also lacks the unit for biovolumes; please add, and also consider to present them as mg/l which would make the axis more readable.

I suggest the authors would also discuss more about the realms of possibility in applying this method to other water bodies and areas in Africa or elsewhere in undeveloped countries. How much external preliminary work, support, expertise and resources are expected to be needed, that was in this study available via the WaSAf project. In this context it would also be important to know about the plans for the continuation of the participatory monitoring in the Lagoon Aghien after the end of the project; how would it succeed in without supporting scientists, smartphones and connections, support etc.

Other comments:

Could the results on quantitative phytoplankton analysis for cyanobacteria be related to the WHO cyanobacteria risk levels?

Abstract (P8): “These data were congruent with those obtained by classical monitoring (water sampling and cell counting) performed over seven months.” This is strongly stated and requires statistical testing; see my earlier comments.

Abstract (P8): the second goal was “to evaluate efficiency”. To my understanding this was not done. Instead it was evaluated how similar or representative the results from citizen observations are when compared to the quantitative biomass results.

P16: Lyngbia -> Lyngbya

P16: I suppose Dolichospermum (Anabaena) filaments were also counted as 100 um counting units. Please add to text.

P16: The HELCOM PEG biovolumes where used to covert cell numbers into biovolumes. HELCOM biovolumes are meant for the Baltic Sea (brackish water) phytoplankton. Could this have some influence of the biovolume estimates?

P21 (1st row): …that there opportunity -> that there is an opportunity

P22: I suppose the method would be suitable for the risk assessment and awareness rising, but not for "traditional" monitoring, because it does not give information on the phytoplankton biomass or species composition.

P22: “Second, citizen monitoring could also permit improved basic knowledge of the dynamics and toxicity of cyanobacterial blooms…” I don’t see how the tested participatory method would help in monitoring cyanobacterial toxicity, as it is only based on visual observations and smartphone photographs.

Reviewer #2: The study reports a pilot study on citizenship science in Africa. It shows the results of a two-year experimentation period collecting data from a limited number of citizens on the development of cyanobacteria in a lagoon in Africa (which provides multiple ecosystem services to citizens) using a smartphone application. The lagoon is used by local populations for various domestic (bathing, washing, cooking, waste management, etc.) and economic (fishing, agriculture, etc.) practices that have an impact and are impacted by the water quality of the lagoon. The highlights and difficulties of this approach are discussed. Noteworthy, most citizen programs happen in developed countries and only few are implemented in developing countries, so I see a great value in this approach, both from the environmental and social point of view. This pilot study showed the low awareness of the populations surrounding the Lagoon concerning the health risks associated with cyanobacterial blooms. The social benefits associated with the citizen monitoring approach were both direct (new learning and understanding of water ecosystem functioning) and indirect (appropriation and translation into conservation actions).

General Comments

Authors should, throughout the manuscript, show that this is a pilot study, a first phase to improve and scale up this project (in its current version, though it is mentioned in the title, the fact that it is a pilot study is not straightforwardly mentioned throughout the manuscript).

The study proposes three goals, here my comments regarding the goals:

Goal 1. Is well developed in the discussion but should be more detailed in the results sections. For example showing all the responses to the questionnaires.

Goal 2. Regarding this goal, that tries to relate the data compiled by the citizens with environmental information, the number of occasions where both types of data were available are too low -4 cases- to state that there is agreement. Authors should early in the draft explain why field data area only available for a 7 month period (they do explain it in the discussion but should be explained before). To be able to address this goal, more field data are needed. Authors can use satellite images to explore the possible relation between cyanobacteria abundance (using Normalized difference chlorophyll ratio, for example, or developing local algorithms) and number of questionnaires reported by citizens.

Goal 3. This goal could be considered as a part of Goal 1 for the three villages assessed (I could not find evidence on how other neighboring populations gained awareness from the experience of the studied villages).

In the introduction I would present first Goal 2 and then Goal 1 as in the results section authors first address Goal 2 and then Goal 1 (or change the order of the results section, describing first the Goal 1).

I understand that this study constitutes a pilot phase of the approach to citizenship monitoring. The fact that this is a pilot study should be presented in all parts of the manuscript: the abstract, introduction, materials and methods, results and discussion. First I would include in the introduction information about the value and need of pilot phases for installing effective programs of citizen monitoring. In the mats and methods section authors need to state that as it is a pilot study only a very limited number of citizens were involved (but that these represented local leaders who had a great respect in their society and a strong disseminating power). I would like to read in the discussion how authors plan to scale up this approach so as to involve a number of citizenships that can represent the population of the villages, and to bridge the limitations to asses Goal 2 (e.g. use satellite images to check relation between environment and society, satellite imagery can be retrieved from the past also). Also I would expect in the discussion how would the scaling up of this project would be and which implications it involves (how many people to involve, expectations: for example when more people get involved the number of responses would markedly increase and the patterns of blooms recorded by the citizens would be more representative of what happens in the water, and to press decision makers to take actions. To bridge the current lack of cross talk between government and citizens, a usual problem in developing countries).

Several parts presented in the Results sections should be presented in the Discussion. I marked them with sticky notes throughout the manuscript.

See below comments on Figures, also see other comments in the text.

Comments about Figures

Figure 1. add a map of Africa with a location of the studied country, add coordinates in the graph. Also add the other 5 villages that are found near the Lagoon.

Figure 2. The results section should provide numbers for these activities. For example. How many reports were answered, how many people participated in the focus groups.

Figure art should be improved, in particular in Fig 3, 5, 6. remove horizontal lines.

Fig. 4. This data correspond to the whole study period? Please clarify in the legend. Put circles with sizes proportional to the number of responses. Add sampling points in this figure, so as to be able to see number of response closest to each sampling point.

Fig 5. For the second period (>sept/2018-2019) I would expect that values in Fig 5 should coincide with those in Fig. 3 (as people were advised to send pictures only if cyanobacteria or other worrying phenomena were observed). Why are they different? Please clarify.

Fig 6. February instead of Fev. In the legend instead of CM = Citizen monitoring, authors should explain that such monitoring correspond only to potential positive observations, as explained in the secondary axis. I could not find in the text or in the legends how a potential case for cyanobacteria was selected (was it based on the information of discoloration of water, or with the photographs, or with both of them?) This information should be included in the article. Add unit of biovolume in Y axis. Why there is only info up to April 2018? If possible, extend to both periods or at least to the end of the first period, so as to increase the number of situations where there is field data and reports from citizens. Bars of CM should have within them the proportion of locations where the record was performed, village or closest location to the sampling point of cyanobacteria counts (S1 to S6) .

Figure 7. The information regarding how this questionnaire was undertaken is absent in the materials and methods section. Please include. Also in the figure state how many people answered this questionnaire.

Supplementary 2. Translate to English the questions asked in the app. Even if the app is French, translate it to English so that the scientific audience can read its content.

Other comments

All other comments are inserted in the manuscript, due to the lack of line numbers in the manuscript I found it easier to write on the draft directly.

Reviewer #3: Author’s designed, implemented and evaluated a project of participatory monitoring of cyanobacteria blooms through the use of a smartphone application, working with the populations of three villages located on the shoreline of a freshwater lagoon located near Abidjan city (Lagoon Aghien in the Ivory Coast). The aim was to analyze the efficiency of such citizen science approach, comparing with classical monitoring, the potential constraints of the approach, and the contribution to citizen awareness about cyanobacteria blooms and their health consequences. Author’s claimed that the participatory approach provided accurate data compared with classical monitoring, and in addition, showed great improvements to the understanding and awareness of local populations regarding water quality issues.

In my opinion, the work is interesting, although the limited data of the classical monitoring make it less relevant than it could have been. It’s a shame that the classical monitoring was unilaterally finished by French institutions. In my personal opinion, there is limited data to account for a solid comparison. In this sense, in my opinion the work is quite limited about the efficacy of the participatory detection of cyanobacteria blooms.

Globally, my recommendation is to consider some major and minor revisions, some of them are important to clarify or improve the manuscript.

Majors

1. The comparison of the participatory approach with the classical one shows some caveats that don’t totally reflect the efficacy of the participatory approach about the detection of cyanobacteria blooms. For example, it appears to be a threshold in the detection of water color changes associated with cyanobacteria blooms. Some smaller blooms were not detected by local people. This issue have to be more addressed on the text.

2. “The importance of water color for the assessment of water quality by inhabitants was seen by us as a positive signal for the implementation of participative monitoring of cyanobacteria blooms.” Results section.

I don’t agree with that. A limitation is that before smartphone application you don't know if they use water color as indicator of water quality. That's a limitation to inform about the awareness associated with monitoring. Please remove or modify accordingly if you have more information previous to the beginning of participatory monitoring.

3. I think that Figure 6 and the comparative analysis of monitoring programs should be improved. It is desirable to have some quantitative measurement of agreement between the classical and the participatory approaches. I suggest some kind of analysis that allow subjective/objective comparisons (e.g. ROC curves), or some kind of association/classification analysis.

Meanwhile, the plot is not very straightforward to understand it, in particular regarding biovolume counts in each station. Related with this, S5 and S6 have the same color? Color codes could also be improved to facilitate identification of stations. Lack of units in y-axis.

Minors

Material and Methods section

4. “As shown by this figure, the participatory monitoring benefited from sociological

investigations conducted within the framework of the WaSAf program…”.

This is not clear in Figure 2. Where is mentioned in Fig. 2 the previous sociological research? Meetings are not only part of sociological research, they can be part of any research. Please modify the text or the Figure accordingly.

5. Figure 2. It is not clear from the figure or the text why the steps are divided in three blocks without a correspondence with the three main titles of the boxes. Why is it? I think that it is confusing, I suggest to edit the figure or to be clearer with that at the legend. For example, Phase 1, preparation, have 2 items and a previous one: 1) first meeting to choice villages, 2) organization of focus groups, 3) meetings and provision of flyers of smartphone application; but the phase was divided in two boxes, the first two items in a box, and the third item on a second box. Why? The same thing with the rest of the phases/boxes.

6. “Meetings and interviews with local community leaders and chiefdom and with

inhabitants of the three villages..”

Please check syntaxes. A comma is lacking somewhere

7. “prospectus (Lagoon Health Bulletin) were also distributed in schools and villages.”

Prospectus were done by scientist or also by local actors, such as sentinels? Please clarify.

8. ” we also conducted qualitative interviews with the three sentinels and other volunteers”.

What kind of interview did author's performed? Informal? Semi-structured? Please clarify the kind of interview and any further details to understand the kind of responses given by interviewers.

9. “The participatory monitoring was organized in two periods…”

Why did authors choose this approach? It was top-down (scientist to local people) or it was discussed or decided with local actors? Can you clarify the approach?

Results section

10. What are the violet points in Figure 4? Please clarify or remove them.

11. Figure 5. To have a better picture of the participatory monitoring of blooms, authors can include within the graph several photos sent by sentinels corresponding to different moments of the monitoring dynamics.

Discussion section

12. “This first citizen project based on the use of a smartphone application for the monitoring of cyanobacterial blooms in Africa shows that there opportunity for such approaches.”

Did authors mean “there is opportunity”? Please correct.

13. Limitations of the program. Authors have limited data to take conclusions about participatory monitoring. Some blooms were not recorded by participants, is not just technical problems regarding geolocalization. This is an important issue and has implications for future monitoring programs.

14. What was the reaction of local people when French institutions unilaterally finished the monitoring? May be it would be interesting to mention at the discussion section.

15. Have sentinels stopped to send reports so far? Which is the long-term sustainability of the program? If author's can expand this observation it would be great.

16. I fully agree with the content of last paragraph but I think this is not a conclusion technically speaking. I suggest that in order to maintain the content in the text, please modify the paragraph and/or add a conclusion relative with the goals of the work.

6. PLOS authors have the option to publish the peer review history of their article (what does this mean?). If published, this will include your full peer review and any attached files.

Reviewer #1: No

Reviewer #2: No

Reviewer #3: No

---

## [Decision Letter · Decision Letter 1]

26 Aug 2020

Can participatory approaches strengthen the monitoring of cyanobacterial blooms in developing countries? Results from a pilot study conducted in the Lagoon Aghien (Ivory Coast)

PONE-D-20-09032R1

Dear Dr. Humbert,

We’re pleased to inform you that your manuscript has been judged scientifically suitable for publication and will be formally accepted for publication once it meets all outstanding technical requirements.

Kind regards,

Cleber Cunha Figueredo, Ph.D.

Academic Editor

PLOS ONE

Additional Editor Comments (optional):

Dear Author,

after two rounds of reviews, your manuscript was really improved and can be published in PLOSONE. However, some minor points were mentioned by reviewers and you should consider to do some changes. Plesase pay attention in the doubts related to the supplementary material.

Best regards,

Cleber Figueredo

Reviewers' comments:

Reviewer's Responses to Questions

**Comments to the Author**

1. If the authors have adequately addressed your comments raised in a previous round of review and you feel that this manuscript is now acceptable for publication, you may indicate that here to bypass the “Comments to the Author” section, enter your conflict of interest statement in the “Confidential to Editor” section, and submit your "Accept" recommendation.

Reviewer #1: All comments have been addressed

Reviewer #2: All comments have been addressed

2. Is the manuscript technically sound, and do the data support the conclusions?

Reviewer #1: (No Response)

Reviewer #2: Yes

3. Has the statistical analysis been performed appropriately and rigorously? 

Reviewer #1: (No Response)

Reviewer #2: Yes

4. Have the authors made all data underlying the findings in their manuscript fully available?

Reviewer #1: (No Response)

Reviewer #2: Yes

5. Is the manuscript presented in an intelligible fashion and written in standard English?

Reviewer #1: (No Response)

Reviewer #2: Yes

6. Review Comments to the Author

Reviewer #1: Review - PONE-D-20-09032R1 Mitroi et al.: ‘Can participatory approaches strengthen the monitoring of cyanobacterial blooms in developing countries? Results from a pilot study conducted in the Lagoon Aghien (Ivory Coast)’

The manuscript has clearly improved during the review. The authors have taken thoroughly into account the comments and concerns I expressed for the earlier version of the manuscript. I find it now a solid and consistent paper where the data supports the conclusions made. I recommend it to be considered to be published as such in PlosONE.

I have only a few minor comments to make:

Page 12, row 280: ‘… only if cyanobacteria or other degradation signs were observed.’ Could this rephrased: ‘…only if cyanobacteria, visible water coloration, or other degradation signs were observed.’

Table 1 on page 28: This new table is informative. However, it is not referred in the text. If I can follow to my understanding the authors’ idea is to include the tablre as a Supplementary Table 1 (ST1), but that is already existing. I recommend to have it as table 1 in the article, not just as Supplementary information.

Reviewer #2: I added minor comments in the pdf, once addressed i believe the paper can be accepted. Authors had included most comments. I cound not see supplementary figs 1 and 2 as only table suppl 1 was retrieved when downloading supp material, maybe it is something with my computer. If other reviewers consider supp figures ok i am also ok

7. PLOS authors have the option to publish the peer review history of their article (what does this mean?). If published, this will include your full peer review and any attached files.

Reviewer #1: No

Reviewer #2: **Yes: **Paula de Tezanos Pinto

---

## [Editor Report · Acceptance letter]

16 Sep 2020

PONE-D-20-09032R1 

Can participatory approaches strengthen the monitoring of cyanobacterial blooms in developing countries? Results from a pilot study conducted in the Lagoon Aghien (Ivory Coast) 

Dear Dr. Humbert:

I'm pleased to inform you that your manuscript has been deemed suitable for publication in PLOS ONE. Congratulations! Your manuscript is now with our production department. 

Kind regards, 

on behalf of

Dr. Cleber Cunha Figueredo 

Academic Editor

PLOS ONE